# The impact of demographic factors and personality traits on nurse compassion fatigue: A cross-sectional analysis

Miao Zhao[1], Min Xie[2]*

1 College of Pharmacy, Chengdu Medical College, Chengdu, Sichuan, China, 2 Department of Cosmetic Plastic and Burn Surgery, The First Affiliated Hospital of Chengdu Medical College, Chengdu, Sichuan, China

* redpinkblue@sina.com

## Abstract

### Background

Compassion fatigue significantly impacts nurses' work efficiency, care quality, and overall well-being. Consequently, the phenomenon of compassion fatigue among nurses has garnered considerable attention. Research from abroad indicates that demographic factors and the Big Five personality traits are closely related to the burnout dimension of compassion fatigue. This insight opens new avenues for studying compassion fatigue in nurses.

### Purpose

This study aims to analyze the current state of compassion fatigue among nurses and its influencing factors. Furthermore, it seeks to explore the correlation between compassion fatigue and personality traits, providing valuable references for alleviating compassion fatigue among nurses.

### Methods

A survey using general information, the Chinese Compassion Fatigue Short Scale, and the brief version of the Chinese Neuroticism Extraversion Openness Five-Factor Inventory was conducted on 332 nurses from a major tertiary hospital.

### Results

The total score for nurse compassion fatigue was 51.83($SD$=22.02). Factors such as years of service, professional title, monthly income, and department were identified as influencing compassion fatigue. The total score for compassion fatigue exhibited a negative correlation with extraversion, openness, agreeableness, and conscientiousness, while it showed a positive correlation with neuroticism. The Big Five personality

**Data availability statement:** All relevant data are within the manuscript and its Supporting Information files.

**Funding:** The author(s) received no specific funding for this work.

**Competing interests:** The authors have declared that no competing interests exist.

traits accounted for 47.9%, 45.2%, and 50.2% of the variance in secondary trauma, burnout, and the overall compassion fatigue score, respectively.

## Conclusions/Implications for Practice

Neuroticism and agreeableness are primary predictors of nurses' compassion fatigue. Nursing managers can implement targeted measures to alleviate compassion fatigue based on the identified influencing factors and the predictive role of the Big Five personality traits.

## Introduction

Compassion fatigue refers to the reduced energy or interest experienced by caregivers in their work, which can lead to burnout and even changes in their values and worldview [1]. This phenomenon is often observed among nurses, as their compassion for patients' suffering can take a toll on their own well-being and work performance. Studies have shown that nurses are prone to compassion fatigue, which not only affects nurses' work efficiency and rescue level, but also affects their physical and mental health [2]. Therefore, the issue of compassion fatigue in the nursing profession has received significant attention. In addition, studies have shown that demographic factors and personality traits, particularly the Big Five, are closely related to the burnout dimension of compassion fatigue among nurses [3]. This provides new insights for the research on compassion fatigue in nursing.

When discussing the relationship between nurses' compassion fatigue and the Big Five personality traits, it is essential to comprehend the specific definitions of the Big Five traits. This model encompasses five core dimensions: neuroticism, extraversion, openness, agreeableness, and conscientiousness [4]. Every dimension affects a person's behavior and mental state. Neuroticism, in particular, is an emotional trait characterized by heightened emotional responses, weak emotional awareness, ineffective emotional coping, and frequent negative feelings [5]. Neurotic individuals are more prone to intense negative emotions like anxiety, depression, and anger, reacting [6]. Their emotional regulation and stress management skills are relatively weak. Nurses with neurotic personalities often experience lower job satisfaction and a higher risk of compassion fatigue, as they are more susceptible to negative emotions and struggle with emotional control and coping strategies [7]. Besides, conscientiousness reflects self-control ability, motivation to achieve, and a sense of responsibility [8]. Agreeableness reflects one's positive attitude toward others, characterized by helpfulness, reliability, empathy, and trust [9]. Extraversion encompasses a preference for socializing, taking initiative, showing enthusiasm and energy, maintaining optimism, and possessing strong communication skills, along with the ability to actively seek solutions in challenging situations [10]. Openness reflects a cognitive style and a willingness to embrace new experiences, characterized by diverse interests, curiosity, and a desire for novelty [11]. By analyzing these five personality traits, we can uncover the potential compassion fatigue experienced by nurses with different traits when confronting various risks.

Research indicates that empathy ability is negatively correlated with the total score of compassion fatigue, and the higher the level of empathy ability, the lower the level of compassion fatigue [12]. Neuroticism represents the negative aspect of personality traits, which is negatively correlated with empathy ability and positively correlated with compassion fatigue [12]. The other four dimensions represent the positive aspect of personality traits, which are positively correlated with empathy ability and negatively correlated with compassion fatigue [13]. Therefore, nurses with high scores on the neuroticism dimension have higher levels of compassion fatigue, while nurses with high scores on the conscientiousness, agreeableness, extraversion, and openness dimensions have lower levels of compassion fatigue. These findings suggest that hospitals should develop personalized intervention measures based on nurses' different demographic factors and personality traits to effectively prevent and alleviate compassion fatigue and ensure the physical and mental health of nurses. Specifically, hospitals can provide nurses with comprehensive support through psychological counseling, stress management courses, continuous education, support systems, and reasonable work arrangements, helping them cope with the emotional challenges in their work and improving job satisfaction and nursing quality.

The study surveyed nurses at a tertiary hospital in Chengdu to understand the status of their compassion fatigue and its influencing factors. It also analyzed the relationship between nurses' compassion fatigue and their demographic characteristics and Big Five personality traits. The results are intended to serve as a reference for hospitals to implement targeted interventions to alleviate nurses' compassion fatigue and safeguard their wellbeing.

## Materials and methods

### Study design

A cross-sectional study was undertaken among a convenience sample of clinical registered nurses from a tertiary hospital in Chengdu, Sichuan, China, using an electronic questionnaire. This study adhered to the Strengthening the Reporting of Observational Studies in Epidemiology (STROBE) guidelines.

### Procedure

The research procedure commenced in March 2023, during which the study concept was formulated and the research protocol developed. Formal ethical approval was obtained in June 2023. Primary data collection using structured electronic questionnaires commenced in January 2024. Upon completion of data collection, comprehensive data cleaning was performed, including identification of missing values, detection of outliers, and exclusion of invalid responses based on predefined validity criteria. Validated datasets were subsequently coded and subjected to statistical analysis. These results formed the basis for manuscript development, which underwent iterative internal review, professional proofreading, and language polishing prior to finalization.

### Sampling and participants

A convenience sampling method was adopted to recruit clinical registered nurses as study participants. The sample size was calculated using G*Power version 3.1.9.7 (Windows XP, Vista, 7, 8, 10 and 11, Heinrich Heine Universität, Düsseldorf, Germany). With an effect size $f$ of 0.25, $\alpha$ error probability of 0.05, and a power (1-$\beta$ error probability) of 0.95, the estimated number of participants was 280. To account for potential invalid questionnaires, the sample size was increased by 10%, leading to a minimum required sample size of 308 participants.

The criteria for participant inclusion were: ① holding a nurse's practice certificate from the People's Republic of China; ② currently employed; ③ work experience ≥ 1 year; ④ no mental illness or consciousness disorders; ⑤ informed consent and voluntary participation in this study. The exclusion criteria were: ① interns, resident nurses, visiting nurses, and non-on-duty personnel (e.g., out-of-office for study, leave); ② those preparing to resign or who have submitted a resignation application.

 

## Ethical considerations

In accordance with the Ethical Considerations, the protocol for this research was approved by the Institutional Review Board at The First Affiliated Hospital of Chengdu Medical College,China. According to Item 2 of Article 32 of the "Ethical Review Measures for Life Sciences and Medical Research Involving Humans" of the People's Republic of China: "Research conducted using anonymized information data that does not cause harm to humans,involve sensitive personal information, or concern commercial interests may be exempted from ethical review."All the participants provided written informed consent prior to completing the survey.They were informed of the purpose of the study and they have the right to withdraw from the study at any time. Their survey data were treated to guarantee confidentiality.

## Instruments

To explore impact of demographic factors and personality traits on nurse compassion fatigue in this sample, the following tools were used: a demographic characteristics questionnaire, a compassion fatigue scale, a five personality factors scale.

**Demographic characteristics questionnaire.** The demographic characteristics questionnaire with basic information, including nurses' sex, age, marital status, education, title, employment type, service years, monthly income, departments, and other general information, was used in this study.

**Compassion fatigue scale.** The Chinese Compassion Fatigue Short Scale (CCF-SS) was used in this study. The English version of the scale was developed by Adams et al through the revision of the Compassion Fatigue Scale-Revised edition [14]. It consists of two dimensions of secondary trauma and job burnout, including 5 items and 8 items respectively, with a total of 13 items, using the 10-point Likert scoring method from 1 (never) to 10 (very frequent), which has acceptable psychometric indicators. This study uses the Chinese version translated by Lou Baona et al. The Cronbach's α coefficients of internal consistency of this scale when applied to medical personnel were 0.83 for secondary trauma, 0.87 for burnout, and 0.90 for total scale.

**Five personality factors scale.** The brief version of the Chinese Neuroticism Extraversion Openness Five-Factor Inventory (CNEO-FFI) was used in this study. Costa and McCrae built on the Big Five personality theory in 1985, based on the factor analysis of the Cattell 16PF scale and their theoretical conception, the Neuroticism Extraversion Openness Personality Inventory (NEO-PI) was compiled to test the five personality factors [15]. Later, it was revised to form NEO-PI-R, and a simplified version of the scale NEO-FFI was introduced at the same time [16]. The Chinese version of NEO-FFI revised by Yang Jian was used in this study, which included five dimensions of neuroticism, extraversion, openness, agreeableness, and conscientiousness. Each dimension was composed of 12 items, and a 5-level score was adopted: 1 score was inconsistent, and 5 score was consistent. The Cronbach's α coefficients of internal consistency in each dimension are: neuroticism 0.81, extraversion 0.76, openness 0.60, agreeableness 0.69, conscientiousness 0.78, so the reliability of the inventory is good.

## Survey methods

The electronic questionnaire was created using the Tencent Questionnaire platform (https://wj.qq.com). Questionnaire administration occurred during regular work shifts. The survey administrator explained the questionnaire's purpose and completion procedures to participating nurses, followed by immediate on-site distribution of the surveys. Participants voluntarily completed the anonymous surveys on personal mobile devices immediately upon receipt, with no compensation provided for participation. The completed questionnaires were then collected immediately. Questionnaires were considered invalid if the personal information was inconsistent, or all options were selected as the same answer.

### Data collection

Data were collected from January to February 2024.360 questionnaires were distributed, and all were collected. In the survey, 28 invalid questionnaires were excluded, leaving 332 valid questionnaires. The overall valid response rate was 92.6%.

### Statistical analysis

After coding, the collected data were analyzed using descriptive statistics, including frequency, percentage, mean, and standard deviation, on the Chinese version of IBM SPSS 23.0 (IBM Inc., Armonk, NY, USA). Independent sample t-test, analysis of variance (LSD method was used for multiple comparisons within the group), Pearson correlation analysis, and multiple hierarchical regression were used for statistical analysis, and $P < .05$ was considered statistically significant.

## Results

### Participant characteristics

The 332 participants involved, with 315 females (94.9%) and 17 males (5.1%). The age distribution was as follows: 188 participants (56.6%) were between 21–30 years old, 115 (34.6%) were between 31–40 years old, and 29 (8.7%) were between 41–50 years old. Marital status: 219 (66.0%) were married, and 113 (34.0%) were unmarried. Educational background: 17 (5.1%) had an associate's degree, 312 (94.0%) had a bachelor's degree, and 3 (0.9%) had a master's degree or higher. Professional title: 37 (11.1%) were nurses, 202 (60.8%) were senior nurses, 83 (25.0%) were supervisor nurse, and 10 (3.0%) were deputy chief nurses or above. Employment status: 64 (19.3%) were permanent staff, and 268 (80.7%) were contract-based nurses. Service years: 51 (15.4%) had 1–3 years, 92 (27.7%) had 4–6 years, 74 (22.3%) had 7–9 years, and 115 (34.6%) had 10 years or more. Monthly income: 52 (15.7%) earned less than 6,000 CNY, 127 (38.3%) earned 6,000–8,000 CNY, 113 (34.0%) earned 8,001–10,000 CNY, and 40 (12.0%) earned more than 10,000 CNY. Nursing departments: 81 (24.4%) worked in internal medicine, 141 (42.5%) in surgery, 57 (17.2%) in the intensive care unit (ICU), and 53 (16.0%) in the operating room.

### Impact of demographic factors on compassion fatigue among clinical nurses

This research examined the influence of demographic variables, such as age, service years, professional title, employment status, income level, and departmental variations, on the compassion fatigue experienced by clinical nurses. Findings indicated that nurses aged 21–30 exhibited significantly higher levels of compassion fatigue compared to their counterparts in other age brackets. Nurses with 4–6 years of service had higher levels of compassion fatigue in all dimensions and overall scores compared to nurses with 7 or more years of service. Similarly, the lower the monthly income of nurses, the higher their level of compassion fatigue. In addition, Supervisor nurses had lower levels of compassion fatigue than senior nurse. The same results are found in general internal medicine nurses, who had relatively low levels of compassion fatigue in all dimensions and overall scores, while surgical nurses, intensive care unit nurses, and operating room nurses had higher overall compassion fatigue scores. To be noteworthy, there were no statistically significant differences in the dimensions of compassion fatigue and overall compassion fatigue between nurses of different sex, marital status, education levels. Therefore, the analysis results of this study indicate that the main demographic factors influencing nurses' compassion fatigue are differences in age, service years, professional title, monthly income, and department (Table 1).

### Correlation between Big Five personality traits and nurses' compassion fatigue

The statistical results for the correlation between the Big Five personality traits and compassion fatigue of clinical nurses in our hospital showed that the nurses' compassion fatigue score was $51.83 \pm 22.02$, with the secondary traumatic stress dimension scoring $17.28 \pm 8.64$ and the burnout dimension scoring $34.55 \pm 14.61$. Within the framework of the Big Five

**Table 1. CCF-SS scores of nurses with different demographic characteristics (x±s, n=332).**

| Items | | Number (n) | Proportion (%) | Secondary trauma | Job burnout | Total compassion fatigue score |
|---|---|---|---|---|---|---|
| **Sex** | ① Male | 17 | 5.1 | 18.06±9.36 | 37.76±15.68 | 55.82±23.70 |
| | ② Female | 315 | 94.9 | 17.24±8.62 | 34.38±14.55 | 51.62±21.94 |
| | t | | | 0.381 | 0.930 | 0.766 |
| | P | | | 0.704 | 0.353 | 0.444 |
| **Marital Status** | ①Married | 219 | 66.0 | 17.07±8.28 | 33.43±14.05 | 50.50±21.08 |
| | ②Unmarried | 113 | 34.0 | 17.69±9.34 | 36.73±15.46 | 54.42±23.62 |
| | t | | | −0.620 | −1.954 | −1.483 |
| | P | | | 0.535 | 0.052 | 0.140 |
| **Age** | ①21~30岁 | 188 | 56.6 | 18.34±9.67 | 37.56±15.62 | 55.90±24.05 |
| | ②31~40岁 | 115 | 34.6 | 16.00±6.89 | 31.20±12.26 | 47.20±17.89 |
| | ③41~50岁 | 29 | 8.7 | 15.48±6.93 | 28.38±11.61 | 43.86±17.30 |
| | F | | | 3.349 | 10.131 | 7.975 |
| | P | | | 0.036 | <0.001 | <0.001 |
| | LSD multiple comparison | | | ①>② | ①>②,③ | ①>②,③ |
| **Service years** | ①1~3年 | 51 | 15.4 | 17.86±9.78 | 35.18±15.88 | 53.04±24.56 |
| | ②4~6年 | 92 | 27.7 | 19.57±10.05 | 41.02±14.86 | 60.59±23.69 |
| | ③7~9年 | 74 | 22.3 | 16.30±7.88 | 32.47±14.01 | 48.77±20.60 |
| | ④≥10年 | 115 | 34.6 | 15.83±6.89 | 30.44±12.38 | 46.27±17.98 |
| | F | | | 3.713 | 10.397 | 8.345 |
| | P | | | 0.012 | <0.001 | <0.001 |
| | LSD multiple comparison | | | ②>③,④ | ②>①,③,④ ①>④ | ②>①,③,④ |
| **Educational background** | ① Associate's degree | 17 | 5.1 | 14.18±7.40 | 28.82±12.03 | 43.00±18.30 |
| | ② Bachelor's degree | 312 | 94.0 | 17.54±8.68 | 34.95±14.71 | 52.49±22.14 |
| | ③ Master degree or above | 3 | 0.9 | 8.00±1.73 | 25.67±9.29 | 33.67±10.41 |
| | F | | | 3.000 | 1.991 | 2.552 |
| | P | | | 0.051 | 0.138 | 0.079 |
| | LSD multiple comparison | | | – | – | – |
| **Title** | ① Nurse | 37 | 11.1 | 17.03±9.359 | 33.65±14.08 | 50.68±22.34 |
| | ② Senior nurse | 202 | 60.8 | 18.51±8.94 | 36.60±14.64 | 55.11±22.25 |
| | ③ Supervisor nurse | 83 | 25.0 | 14.71±7.23 | 30.28±14.67 | 44.99±20.88 |
| | ④ Deputy chief nurse or above | 10 | 3.0 | 14.70±5.76 | 32.00±5.98 | 46.70±9.75 |
| | F | | | 4.235 | 3.951 | 4.525 |
| | P | | | 0.006 | 0.009 | 0.004 |
| | LSD multiple comparison | | | ②>③ | ②>③ | ②>③ |
| **Employment** | Permanent staff | 64 | 19.3 | 15.31±7.65 | 33.92±15.65 | 49.23±22.00 |
| | Contract-based nurse | 268 | 80.7 | 17.75±8.81 | 34.71±14.37 | 52.46±22.02 |
| | t | | | −2.036 | −0.385 | −1.052 |
| | P | | | 0.043 | 0.700 | 0.294 |
| **Monthly income (Yuan)** | ①≤6000 | 52 | 15.7 | 18.38±10.61 | 36.83±15.85 | 55.21±25.43 |
| | ②6001~8000 | 127 | 38.3 | 17.99±9.17 | 36.54±15.65 | 54.54±23.75 |
| | ③8001~10000 | 113 | 34.0 | 16.86±7.02 | 33.00±12.46 | 49.86±17.76 |
| | ④>10000 | 40 | 12.0 | 14.78±7.97 | 29.68±13.93 | 44.45±20.82 |
| | F | | | 1.972 | 3.181 | 2.897 |
| | P | | | 0.148 | 0.024 | 0.035 |
| | LSD multiple comparison | | | ①,②>④ | ①,②>④ | ①,②>④ |

*(Continued)*

Table 1. (Continued)

| Items | | Number (n) | Proportion (%) | Secondary trauma | Job burnout | Total compassion fatigue score |
|---|---|---|---|---|---|---|
| **Department** | ① Internal medicine | 81 | 24.4 | 14.74±7.16 | 29.80±13.07 | 44.54±18.86 |
| | ②Surgery division | 141 | 42.5 | 17.65±9.50 | 35.88±15.28 | 53.52±23.73 |
| | ③ Intensive care unit | 57 | 17.2 | 20.63±9.70 | 38.26±16.01 | 58.89±24.35 |
| | ④ Operating room | 53 | 16.0 | 16.58±5.41 | 34.30±11.62 | 50.89±15.41 |
| | *F* | | | 5.609 | 4.622 | 5.435 |
| | *P* | | | 0.001 | 0.004 | 0.001 |
| | LSD multiple comparison | | | ②,③,④>① ③>②,④ | ②,③>① | ②,③>① |

personality dimensions, neuroticism exhibited a markedly positive correlation with all facets of compassion fatigue as well as the overall compassion fatigue score. In contrast, the traits of conscientiousness, agreeableness, extraversion, and openness demonstrated highly negative correlations with each component of compassion fatigue and the aggregate compassion fatigue score. This is consistent with the findings of Namikawa et al. [17] and Decuyper et al. [18]. The scores of the Big Five personality traits of nurses from low to high are neuroticism, openness, extraversion, agreeableness, and conscientiousness (Table 2). Pearson correlation analysis showed that the various dimensions of compassion fatigue and the total compassion fatigue score of nurses in our hospital were significantly positively correlated with the neuroticism dimension of the Big Five personality traits, and significantly negatively correlated with the other dimensions of the Big Five personality traits (Table 3).

Furthermore, we employed a hierarchical multiple regression analysis, treating the dimensions and overall score of compassion fatigue as dependent variables. Demographic characteristics were incorporated as first-level independent variables, while the dimensions of the Big Five personality traits served as second-level independent variables. The findings revealed that, after controlling for demographic factors, the Big Five personality traits accounted for 47.9%, 45.2%, and 50.2% of the variance in secondary trauma, burnout, and the overall compassion fatigue score, respectively. (**Table 4**). The findings of this research reveal that neuroticism and agreeableness, as delineated within the Big Five personality framework, serve as the predominant determinants of compassion fatigue among nurses. Specifically, individuals exhibiting higher levels of neuroticism are more susceptible to experiencing compassion fatigue, whereas those displaying greater agreeableness are less prone to such outcomes. Consequently, the multiple regression analysis incorporating the Big Five personality traits effectively elucidates the variance in both the individual dimensions and the overall scores of compassion fatigue in nursing professionals, thereby facilitating accurate predictions regarding their compassion fatigue status.

## Discussion

Nurse compassion fatigue significantly correlates with demographic factors (age, service years, title, income, department) and Big Five personality traits. Our hospital's mean score (51.83±22.02) was significantly higher than Zhang et al.'s pre-COVID-19-pandemic finding [19] but lower than Yi et al.'s during-pandemic result [20], reflecting a transitional post-pandemic status. During the pandemic, rigorous infection protocols compelled nurses to balance clinical duties with safety compliance, escalating workloads and intensifying tension between empathy toward patients and safety mandates. Hospital staffing shortages further exacerbated compassion fatigue risks. Yi et al. [20] reported heightened compassion fatigue in nurses during the pandemic compared to pre-pandemic levels. Our findings further indicate this effect persisted beyond the pandemic's conclusion.

Multiple demographic variables significantly influenced nurses' compassion fatigue, particularly years of service, professional titles, and monthly income. Early-career nurses with 1–3 years (27.1% of 21–30 cohort) and 4–6 years (48.9%)

**Table 2. CCF-SS and CNEO-FFI scores of nurses.**

| Items | Dimension | Terms | Score range | Dimension score | Term average score |
|---|---|---|---|---|---|
| **Compassion fatigue** | Secondary trauma | 5 | 5~43 | 17.28±8.64 | 3.46±1.73 |
| | Job burnout | 8 | 8~73 | 34.55±14.61 | 4.32±1.83 |
| | Total compassion fatigue score | 13 | 13~116 | 51.83±22.02 | 4.00±1.69 |
| **Big Five Personality** | Neuroticism | 12 | 12~56 | 35.03±7.86 | 2.92±0.65 |
| | Extraversion | 12 | 20~60 | 37.27±6.59 | 3.12±0.55 |
| | Openness | 12 | 26~52 | 37.36±3.97 | 3.11±0.33 |
| | Agreeableness | 12 | 28~54 | 42.14±5.06 | 3.51±0.42 |
| | Conscientiousness | 12 | 23~60 | 42.81±7.28 | 3.57±0.61 |

**Table 3. Correlation analysis between nurses' CCF-SS scores and CNEO-FFI scores (R-value).**

| | Neuroticism | Extraversion | Openness | Agreeableness | Conscientiousness |
|---|---|---|---|---|---|
| **Secondary trauma** | 0.593** | −0.387** | −0.270** | −0.611** | −0.372** |
| **Job burnout** | 0.682** | −0.507** | −0.288** | −0.510** | −0.426** |
| **Total compassion fatigue score** | 0.685** | −0.488** | −0.297** | −0.578** | −0.429** |

*$P<0.05$, **$P<0.01$.

of experience reported higher fatigue than those with 7–9 years, reflecting skill-acquisition pressures versus mid-career work-life strains. Title disparities revealed junior nurses' vulnerability to promotion-related demands, while intermediate/senior titles conferred experience-based psychological resilience. Critically, lower monthly income directly correlated with elevated fatigue (aligning with Tran et al. [21]), as financial security enhanced career optimism and coping capacity.

Significant inter-departmental variance in compassion fatigue emerged(aligning with Yi et al. [20]), rooted fundamentally in differential patient profiles, clinical conditions, and treatment modalities across departments. ICU nurses exhibited elevated levels (aligning with Tran et al. [21] and Xie et al. [22]), whereas Internal Medicine reported the lowest scores—directly resulting from sustained nurse-patient interactions with predominantly chronic-care patients, which foster robust therapeutic rapport that mitigates compassion fatigue. The elevated compassion fatigue among nurses in high-risk departments (Surgery/ICU/OR) relates to several factors: heightened exposure to urgent and critical care scenarios with associated substantial responsibilities and performance demands [23]; work mode characterized by rapid tempo, sustained pressure, and emotional labor, resulting in cumulative physical-mental fatigue; and restrictive environmental conditions featuring prolonged standing with limited interpersonal interaction/emotional expression, which contribute to intensified professional burnout [24,25].

However, this investigation revealed no statistically significant variations in the dimensions of compassion fatigue or the overall levels of compassion fatigue between permanently employed nurses and those on contract ($P=0.294$), attributable to uniform compensation structures and identical professional development policies across employment types.

Furthermore, the collective personality profile demonstrated the highest levels of conscientiousness, reflecting nurses' diligence and patient-centered dedication, followed sequentially by agreeableness, extraversion, and openness, whereas neuroticism registered the lowest scores. Significant associations emerged between Big Five personality traits and compassion fatigue, with neuroticism showing a strong positive correlation. This relationship originates from neurotic individuals' tendency toward negative cognition and avoidant coping strategies, which contribute to emotional dysregulation and psychological distress, consequently increasing susceptibility to compassion fatigue. In contrast, other traits exhibited negative correlations, among which agreeableness demonstrated the strongest absolute association magnitude. Heightened

**Table 4. Multivariate hierarchical regression analysis of nurses' CCF-SS scores and the Big Five personality traits (β value).**

| Variables | Secondary trauma | | Job burnout | | Total compassion fatigue score | |
|---|---|---|---|---|---|---|
| | Step 1 | Step 2 | Step 1 | Step 2 | Step 1 | Step 2 |
| **Controlled variable** | | | | | | |
| **Sex** | −0.030 | 0.010 | −0.051 | −0.007 | −0.046 | 0.001 |
| **Age** | −0.033 | 0.033 | −0.190* | −0.124 | −0.139 | −0.070 |
| **Marriage status** | −0.113 | −0.159** | −0.024 | −0.077 | −0.060 | −0.114* |
| **Academic degree** | 0.031 | 0.060 | 0.005 | 0.004 | 0.009 | 0.026 |
| **Title** | −0.043 | −0.151** | 0.061 | −0.045 | 0.024 | −0.089 |
| **Employment** | 0.104 | 0.119** | 0.007 | 0.000 | 0.046 | 0.047 |
| **Service years** | −0.093 | 0.048 | −0.082 | 0.078 | −0.090 | 0.070 |
| **Monthly income** | −0.062 | −0.128** | −0.053 | −0.137** | −0.059 | −0.141** |
| **Department** | 0.020 | 0.038 | 0.030 | 0.026 | 0.028 | 0.032 |
| **Big Five Personality** | | | | | | |
| **Neuroticism** | | 0.417** | | 0.531** | | 0.516** |
| **Extraversion** | | 0.071 | | −0.052 | | −0.007 |
| **Openness** | | −0.058 | | −0.034 | | −0.045 |
| **Agreeableness** | | −0.439** | | −0.186** | | −0.295** |
| **Conscientiousness** | | 0.050 | | 0.020 | | 0.033 |
| *F* value | 1.514 | 24.499** | 2.486** | 24.262** | 2.045** | 28.370** |
| R² | 0.041 | 0.520 | 0.065 | 0.517 | 0.054 | 0.556 |
| △R² | | 0.479 | | 0.452 | | 0.502 |

*$P < 0.05$, **$P < 0.01$.

agreeableness served as a protective factor, enhancing nurses' capacity for empathic engagement, therapeutic communication, and professional identification. These competencies collectively enhance nurse-patient rapport, alleviate nurses' emotional exhaustion, and reduce their susceptibility to compassion fatigue.

It should be noted that compassion fatigue not only affects the physical and mental health of nurses, but also has a negative impact on patient care. Nurses effected by compassion fatigue experience a decreased ability to feel empathy and hence lack meaning in their work,which results in substandard care [26,27]. When patients sense the impact of compassion fatigue, they question the quality and appropriateness of care, which in turn escalates patient stress and may ultimately lead to a poor patient's prognosis [28].

Leveraging the Big Five model's predictive capacity for nurses' compassion fatigue enables optimized clinical management through three synergistic approaches: (1) personality-informed staffing that prioritizes assigning high-agreeableness nurses to roles requiring extensive patient interaction, leveraging their natural resistance to compassion fatigue;(2) trait development initiatives harnessing personality malleability via psychological interventions to cultivate adaptive characteristics and stimulate professional engagement; and(3) proactive risk mitigation identifying high-risk cohorts through trait profiling for targeted support—collectively enhancing workforce stability, refining clinical competencies, and maximizing human resource utilization.

## Limitations

This study has some limitations. First, the cross-sectional design, while efficient for capturing prevalence data, inherently precludes causal inference between personality traits and compassion fatigue due to simultaneous exposure-outcome measurement. Second, convenience sampling from a single western Chinese hospital risks selection bias,

limiting generalizability to broader nursing populations given interregional variations in management systems, workload intensity, and clinical environments. Third, although generational characteristics constitute significant demographic confounders, the absence of empirically established cohorts specific to China's sociocultural context precluded their analysis. Fourth, exclusive reliance on self-reported instruments introduces response bias, compounded by suboptimal internal consistency (Cronbach's α = 0.60) in the Openness subscale, collectively potentially compromising measurement precision. Finally, analytical scope was constrained to hierarchical regression without structural equation modeling to elucidate complex trait interactions.

## Conclusions

In summary, the phenomenon of compassion fatigue among nurses necessitates not only individual intervention but also collective recognition and support from both organizational structures and society at large. To enhance nurses' engagement in their professional roles, nursing administrators should prioritize the determinants of compassion fatigue and implement targeted strategies to mitigate its effects. For instance, nursing leaders can leverage the predictive capacity of personality traits regarding compassion fatigue, perform thorough evaluations of nurses' personality profiles, and adopt personalized management approaches through a multifaceted strategy to effectively alleviate compassion fatigue. This, in turn, would elevate nurses' job satisfaction and the quality of nursing care, thereby facilitating a more efficient and effective delivery of services to patients within the nursing profession.

## Supporting information

**S1 Data. The Supporting information is a dataset generated through numerical coding of the questionnaire survey data.** (XLS)

## Acknowledgments

The authors acknowledge the contributions of all the participants.

## Author contributions

**Conceptualization:** Min Xie.

**Data curation:** Miao Zhao.

**Formal analysis:** Miao Zhao.

**Funding acquisition:** Min Xie.

**Investigation:** Miao Zhao, Min Xie.

**Methodology:** Miao Zhao, Min Xie.

**Resources:** Min Xie.

**Supervision:** Min Xie.

**Writing – original draft:** Miao Zhao.

**Writing – review & editing:** Min Xie.

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
