## [Decision Letter · Decision Letter 0]

Dear Dr. Xie,

We look forward to receiving your revised manuscript.

Kind regards,

Majed Sulaiman Alamri, PhD

Academic Editor

PLOS ONE

Journal Requirements:

The URL of TUM datasets in the article is https://cvg.cit.tum.de/data/datasets/rgbd-dataset/download”.

5. Please amend your list of authors on the manuscript to ensure that each author is linked to an affiliation. Authors’ affiliations should reflect the institution where the work was done (if authors moved subsequently, you can also list the new affiliation stating “current affiliation:….” as necessary).

6. We note that the grant information you provided in the ‘Funding Information’ and ‘Financial Disclosure’ sections do not match.

7. We note that Figure 1, 2, 5, and 8 includes an image of a [patient / participant / in the study].

8. Please include your full ethics statement in the ‘Methods’ section of your manuscript file. In your statement, please include the full name of the IRB or ethics committee who approved or waived your study, as well as whether or not you obtained informed written or verbal consent. If consent was waived for your study, please include this information in your statement as well.

9. We note that your Data Availability Statement is currently as follows: All relevant data are within the manuscript and its Supporting Information files.

Reviewers' comments:

Reviewer's Responses to Questions

**Comments to the Author**

1. Is the manuscript technically sound, and do the data support the conclusions?

Reviewer #1: Yes

Reviewer #2: Yes

2. Has the statistical analysis been performed appropriately and rigorously?

Reviewer #1: Yes

Reviewer #2: Yes

3. Have the authors made all data underlying the findings in their manuscript fully available?

Reviewer #1: Yes

Reviewer #2: Yes

4. Is the manuscript presented in an intelligible fashion and written in standard English?

Reviewer #1: Yes

Reviewer #2: Yes

Reviewer #1: Dear Authors,

Your article entitled “The impact of demographic factors and personality traits on nurse compassion fatigue: A cross-sectional analysis” has been edited correctly. The methodology and pragmatics are satisfactory. The analyses are reported correctly and allow for replication. The descriptions are clear and understandable. I suggest adding a subchapter entitled “Procedure” to the Methodology section, describing the research procedure from data collection to the present report. “Participant Characteristics” should be moved to the Methodology section. Please also add section and subsection numbering to make the paper easier to read for potential readers. Furthermore, in terms of content, when discussing demographic factors, the specific characteristics of contemporary generations should be highlighted. Although this is not your leading variable, it falls within the broadly understood confounding variable. Therefore, I suggest adding a mention of contemporary generations in relation to demographic factors in the introduction. Consider referring to doi: 10.3389/fspor.2024.1416154

Thank You!

Reviewer #2: The introduction addresses a relevant and timely issue compassion fatigue among nurses and highlights the potential influence of personality traits and empathy. However, while the statement about the negative correlation between empathy and compassion fatigue is valuable, the broader implications of this relationship are not clearly discussed. I recommend elaborating on how reduced compassion fatigue particularly through improved empathy and personality awareness can positively impact not only nurse well-being but also patient care outcomes.

Method section

The overall structure of the Methods section is clear; however, some areas need further improvement:

It would strengthen the paper to briefly explain why a cross-sectional design and convenience sampling were chosen and to acknowledge their limitations.

The response rate of 92.6% is high, which is positive. However, it would be helpful to clarify how the sample size was determined for example, was a power calculation performed?

Regarding the participants:

Did they complete the survey during working hours or on their own time?

Was participation completely voluntary and anonymous?

The use of the Chinese versions of the CCF-SS and CNEO-FFI is appropriate.

You’ve included reliability scores, which is good. However, the openness subscale shows a relatively low Cronbach’s alpha (0.60), so it may be worth briefly noting this as a limitation.

Discussion

The Discussion section presents relevant insights and draws helpful connections to prior research. However, I recommend revising this section for clarity, focus, and conciseness. While the content is valuable, the current version is too lengthy and at times repetitive, with detailed role descriptions that could be summarized. It would also be helpful to include a brief reflection on how compassion fatigue may impact patient care, and conclude with a clear summary of key findings and their practical implications.

**Do you want your identity to be public for this peer review?** For information about this choice, including consent withdrawal, please see our Privacy Policy

Reviewer #1: No

Reviewer #2: No

---

## [Author Response · Author response to Decision Letter 1]

27 Jun 2025

Part 1: Response to Academic Editor

Dear Dr. Alamri

We sincerely appreciate your editorial guidance and the opportunity to revise our manuscript. We have thoroughly addressed all points raised during the review process:

1. Manuscript Formatting & Language Polishing

The manuscript has been strictly reformatted according to the PLOS ONE style templates. Additionally, comprehensive linguistic revisions were performed to: (1)correct grammatical, typographical and ambiguity errors ;(2)ensure clarity and compliance with PLOS ONE's language standards ;(3)Validate terminology consistency. All modifications related to formatting and language refinement are highlighted in yellow in the 'Revised Manuscript with Track Changes'.

2.ORCID Validation

The corresponding author's ORCID (0000-0002-1876-7993) has been validated in Editorial Manager.

3.Code Sharing

No author-generated code was used in this study

4.Data Availability

The minimal dataset required to replicate our findings has been uploaded as Supporting Information files in the online submission system.

5.Author Affiliation

All authors are now linked to their respective affiliations in the manuscript.

6.Funding Statement

The authors received no specific funding for this work. This declaration has been entered in the online submission system and updated in the Funding Information section.

7.Image Consent

This study contains no figures depicting patients/participants, thus no consent forms are required.

8.Ethics Statement

The full ethics statement has been added to the Methods section (Highlighted in Yellow in Page 5-6, Line 131-140). Scanned copies of both the Ethical Review Opinion Form and its English translation have also been uploaded to the online submission system.

Part 2: Response to Reviewers

Reviewer #1

Comment 1. I suggest adding a subchapter entitled “Procedure” to the Methodology section, describing the research procedure from data collection to the present report.

Response: Thank you very much. We have revised this as suggested in the Methodology section (Highlighted in Yellow in Page 4-5, Line 104-114).

Comment 2. “Participant Characteristics” should be moved to the Methodology section.

Response: Thank you very much. While we appreciate the rationale underlying this recommendation, the disciplinary standards for reporting cross-sectional data—as demonstrated in the referenced PLOS ONE study (doi:10.1371/journal.pone.0324456)—support the inclusion of “Participant Characteristics”within Results sections. To meet PLOS ONE's style requirements, we maintain the original placement.

Comment 3. Please also add section and subsection numbering to make the paper easier to read for potential readers.

Response: Thank you very much.Aligning with PLOS ONE's style guidelines for hierarchical section labeling and the explicit formatting precedent established in the referenced PLOS ONE study�doi:10.1371/journal.pone.0323867�, we have systematically numbered all subsection headings throughout the Results section.(Highlighted in Yellow in Page 7-12, Line 197,213,236).

Comment 4. Furthermore, in terms of content, when discussing demographic factors, the specific characteristics of contemporary generations should be highlighted. Although this is not your leading variable, it falls within the broadly understood confounding variable. Therefore, I suggest adding a mention of contemporary generations in relation to demographic factors in the introduction. Consider referring to doi: 10.3389/fspor.2024.1416154

Response: Thank you very much.We acknowledge the conceptual merit of incorporating generational dimensions. However, cross-cultural validation gaps impede direct application of Western generational taxonomies to China's demographic landscape. Current scholarship lacks operationally valid generational cohorts specific to China's socioeconomic evolution. Consequently, we have deliberately reserved this aspect for critical examination in the Limitations section(Highlighted in Yellow in Page 17, Line 362-364).

Reviewer #2

Comment 1. The overall structure of the Methods section is clear; however, some areas need further improvement:It would strengthen the paper to briefly explain why a cross-sectional design and convenience sampling were chosen and to acknowledge their limitations.

Response: Thank you very much.We selected a cross-sectional design to efficiently capture how demographic factors and personality traits collectively influence compassion fatigue at a critical post-COVID-19-pandemic juncture, prioritizing timely evidence generation over longitudinal causality assessment. Convenience sampling maximized participation feasibility given nurses' unpredictable shift patterns amid hospital COVID-19 recovery initiatives. We have revised this as suggested in the Limitations section (Highlighted in Yellow in Page 17, Line 356-362).

Comment 2. The response rate of 92.6% is high, which is positive. However, it would be helpful to clarify how the sample size was determined for example, was a power calculation performed?

Response: Thank you very much.The sample size was calculated using G*Power version 3.1.9.7. We have revised this as suggested in the Methods section (Highlighted in Yellow in Page 5, Line 117-122).

Comment 3. Regarding the participants:Did they complete the survey during working hours or on their own time?Was participation completely voluntary and anonymous?

Response: Thank you very much.Questionnaire administration occurred during regular work shifts. Participants voluntarily completed the anonymous surveys on personal mobile devices immediately upon receipt, with no compensation provided for participation. We have revised this as suggested in the Methods section (Highlighted in Yellow in Page 7, Line 176-181).

Comment 5.The use of the Chinese versions of the CCF-SS and CNEO-FFI is appropriate.You’ve included reliability scores, which is good. However, the openness subscale shows a relatively low Cronbach’s alpha (0.60), so it may be worth briefly noting this as a limitation.

Response: Thank you very much. We have revised this as suggested in the Limitations section (Highlighted in Yellow in Page 17, Line 364-367).

Comment 6.The Discussion section presents relevant insights and draws helpful connections to prior research. However, I recommend revising this section for clarity, focus, and conciseness. While the content is valuable, the current version is too lengthy and at times repetitive, with detailed role descriptions that could be summarized. It would also be helpful to include a brief reflection on how compassion fatigue may impact patient care, and conclude with a clear summary of key findings and their practical implications.

Response: Thank you very much. We have revised the Discussion section to enhance conciseness while preserving its original structure and key insights. Specifically: �1�Streamlined each paragraph to eliminate redundancies.Reduced the word count from 1,412 to 662 �2�Added analysis on compassion fatigue's impact on patient care (Highlighted in Yellow in Page 16, Line 338-344) ��3�Repositioned references 23 and 24 to ensure contextual coherence(Highlighted in Yellow in Page 19, Line 452-457) ��4�Incorporated three new references (26-28) supporting clinical implications (Highlighted in Yellow in Page 19, Line 460-467).

We believe these revisions significantly strengthen the manuscript’s rigor and clinical relevance. Thank you for your time and consideration.

---

## [Decision Letter · Decision Letter 1]

The impact of demographic factors and personality traits on nurse compassion fatigue: A cross-sectional analysis

PONE-D-25-23186R1

Dear Dr. Xie,

We’re pleased to inform you that your manuscript has been judged scientifically suitable for publication and will be formally accepted for publication once it meets all outstanding technical requirements.

Kind regards,

Majed Sulaiman Alamri, PhD

Academic Editor

PLOS ONE

Additional Editor Comments (optional):

Reviewers' comments:

Reviewer's Responses to Questions

**Comments to the Author**

Reviewer #1: All comments have been addressed

Reviewer #2: All comments have been addressed

2. Is the manuscript technically sound, and do the data support the conclusions?

Reviewer #1: Yes

Reviewer #2: Yes

3. Has the statistical analysis been performed appropriately and rigorously?

Reviewer #1: Yes

Reviewer #2: Yes

4. Have the authors made all data underlying the findings in their manuscript fully available?

Reviewer #1: Yes

Reviewer #2: Yes

5. Is the manuscript presented in an intelligible fashion and written in standard English?

Reviewer #1: Yes

Reviewer #2: Yes

Reviewer #1: Dear Authors,

Thank you for your answers and corrections. The article has significantly improved in quality and I recommend it for publication.

Reviewer #2: (No Response)

**Do you want your identity to be public for this peer review?** For information about this choice, including consent withdrawal, please see our Privacy Policy

Reviewer #1: No

Reviewer #2: **Yes: ** Haifa Albeladi

---

## [Editor Report · Acceptance letter]

PONE-D-25-23186R1

PLOS ONE

Dear Dr. Xie,

I'm pleased to inform you that your manuscript has been deemed suitable for publication in PLOS ONE. Congratulations! Your manuscript is now being handed over to our production team.

Kind regards,

on behalf of

Prof. Majed Sulaiman Alamri

Academic Editor

PLOS ONE